

# Hydrogen sulphide alleviates *Fusarium* Head Blight in wheat seedlings

Yuanyuan Yao[1,2], Wenjie Kan[1], Pengfei Su[1,2], Yan Zhu[1,2], Wenling Zhong[1,2], Jinfeng Xi[1,2], Dacheng Wang[1], Caiguo Tang[1] and Lifang Wu[1,2,3]

[1] Hefei Institutes of Physical Science, Chinese Academy of Sciences, Hefei, China
[2] University of Science and Technology of China, Hefei, China
[3] Zhongke Taihe Experimental Station, Taihe, China

## ABSTRACT

Hydrogen sulphide ($H_2S$), a crucial gas signal molecule, has been reported to be involved in various processes related to development and adversity responses in plants. However, the effects and regulatory mechanism of $H_2S$ in controlling *Fusarium* head blight (FHB) in wheat have not been clarified. In this study, we first reported that $H_2S$ released by low concentrations of sodium hydrosulphide (NaHS) could significantly alleviate the FHB symptoms caused by *Fusarium graminearum* (*F. graminearum*) in wheat. We also used coleoptile inoculation to investigate the related physiological and molecular mechanism. The results revealed that FHB resistance was strongly enhanced by the $H_2S$ released by NaHS, and 0.3 mM was confirmed as the optimal concentration. $H_2S$ treatment dramatically reduced the levels of hydrogen peroxide ($H_2O_2$) and malondialdehyde (MDA) while enhancing the activities of antioxidant enzymes. Meanwhile, the relative expressions levels of defence-related genes, including *PR1.1*, *PR2*, *PR3*, and *PR4*, were all dramatically upregulated. Our results also showed that $H_2S$ was toxic to *F. graminearum* by inhibiting mycelial growth and spore germination. Taken together, the findings demonstrated the potential value of $H_2S$ in mitigating the adverse effects induced by *F. graminearum* and advanced the current knowledge regarding the molecular mechanisms in wheat.

# INTRODUCTION

Wheat is an important grain crop whose production safety has global economic significance. However, many factors, including abiotic and biotic stresses, can reduce wheat yield and quality (*Ghimire et al., 2020*). Wheat *Fusarium* head blight (FHB), is a devastating disease in wheat that is mainly caused by *Fusarium graminearum* (*F. graminearum*) (*Qiu, Xu & Shi, 2019*) and is also known as "wheat cancer". In the growing season, it can infect the spike, stem base, and root of the grain under long-term wet and warm environments (*McMullen et al., 2012*). FHB presents as bleaching or necrotic browning of spike tissue, ultimately producing shrunken or discoloured grains (*Buhrow et al., 2016*). The resulting loss of production and quality causes significant economic losses worldwide every year (*McMullen, Jones & Gallenberg, 1997*).

Corresponding authors
Caiguo Tang, cgtang@ipp.ac.cn
Lifang Wu, lfwu@ipp.ac.cn

Various methods have been proposed to control wheat FHB, which mainly involve breeding FHB-resistant cultivars and application of chemical fungicides. Although tens of thousands of wheat accessions have been screened worldwide, no completely resistant germplasm has been found (*Wang et al., 2020*). Currently, more than 200 FHB resistance QTLs have been mapped, whereas only the genes *Fhb1* and *Fhb7* had been cloned, and are expected to provide genetic resources for global wheat production and food safety (*Ghimire et al., 2020*). *Fhb1* from Sumai 3 has been recognised to offer the best chance of enhancing the resistance of FHB, and had been widely used as a parent in hybrid crosses (*Su et al., 2019*). Nevertheless, its resistance mechanism remains elusive (*Hao et al., 2020*). *Wang et al. (2020)* found that *Fhb7* from *Thinopyrum elongatum* conferred resistance to FHB without a yield penalty in wheat. Notably, since resistance, agronomic traits, and quality traits are all controlled by multiple genes, the application of new resistance candidates in production requires comprehensive evaluation (*Jia et al., 2018*). In addition, the use of fungicides has been widely attempted to control FHB (*Mesterházy, Bartók & Lamper, 2003*). However, this approach has been shown various deficiencies, including inadequate effectiveness and the widespread existence of resistance to some fungicides (*Duan et al., 2014*). The first carbendazim (MBC)-resistant ($MBC^R$) field isolate of *F. graminearum* was discovered by *Zhou, Ye & Liu (1994)*. In addition, *Becher et al. (2010)* reported the emergence of quantitative azole resistance in *F. graminearum*. Thus, development of new and efficient methods for pathogen control is extremely important.

$H_2S$, which plays various roles in regulating plant physiology, has recently been recognised as the third gaseous messenger after nitric oxide (NO) and carbon monoxide (CO) (*Wang, 2002*). The high permeability of small gaseous molecules facilitates signal transduction through the membrane (*Wang, 2002*). Available experimental information shows that exogenous applications of $H_2S$ can alleviate the negative effects of numerous abiotic stresses, including salinity (*Deng et al., 2016*), drought (*Ma et al., 2016*), heavy metal (*Zhang et al., 2008*; *Zhang et al., 2010a*; *Zhang et al., 2010b*), temperature (*Yang et al., 2016*), and water (*Shan et al., 2011*) stresses. However, there were few studies on the regulation of biotic stress resistance by $H_2S$ in plants.

In this study, we first reported that exogenous application of low concentrations of $H_2S$ could effectively control FHB in wheat. Then, by evaluating the $H_2O_2$ and MDA contents, activities of antioxidant enzymes, relative expressions levels of defence-related genes, and antibacterial effects, we performed a preliminary exploration of the underlying mechanisms by which $H_2S$ alleviated FHB in wheat seedlings. $H_2S$ is easy to obtain, inexpensive and shows multiple-effects, and exogenous $H_2S$ treatment may be a prospective strategy to control wheat FHB.

## MATERIALS & METHODS

### Regents and media

Sodium hydrosulphide (NaHS) (Sinopharm Chemical Reagent Co., Ltd., Shanghai, China) served as the donor for $H_2S$. Carboxymethyl cellulose (CMC) medium (15 g/L carboxymethyl cellulose, 1 g/L yeast extract, and 1 g/L $NH_4NO_3$) was used for liquid cultures of fungi.

## Plant materials and treatments

The local virulent *F. graminearum* Schw., was kindly provided by Professor Xiue Wang, Nanjing Agricultural University, was used for FHB evaluation. *F. graminearum* inoculum was prepared in accordance with the method described by *Buhrow et al. (2016)*. *F. graminearum* conidia collected from PDA agar plates were transferred into liquid medium (CMC) and cultivated at 28 °C for 5 d with shaking at 200 rpm. Then conidial suspension was filtrated through six layers of gauze and centrifuged at 6500 rpm for 6 min, and the spore concentration was adjusted to $3.2 \times 10^7$ conidia/mL (0.1% Tween 80) with a haemocytometer and light microscopy.

Two-day-old seedlings of the FHB-susceptible wheat variety 'Bainong 207' were tested for coleoptile infection analysis by using the method described by *Wu et al. (2005)* with minor alterations. Two days post-sowing (dps), the top 1–2 mm of the coleoptiles was cut off, and 2 µL of *F. graminearum* inoculum was added to the top of the remaining seedlings. Coleoptiles inoculated with 0.1% Tween-80 served as controls. Then, the inoculated seedlings were grown in controlled environmental conditions (temperature: 25 °C; photoperiod: 16 h light/8 h dark; relative humidity: 95%).

NaHS was used as the $H_2S$ donor according to the method described by *Hosoki, Matsuki & Kimura (1997)*. In order not to affect the growth of plants, we first determined the application concentration range of NaHS. Two-day-old wheat seedlings were culture in 0.0 (control), 0.1, 0.3, 0.5, and 1.0 mM NaHS respectively for 3 days, then the growth index was investigated on 9 dps (days post sowing). To explore the potential functions of $H_2S$ in FHB resistance of wheat, the seedlings were pre-treated with 0, 0.1, 0.3, and 0.5 mmol/L (mM) NaHS for 6 h; exposed to *F. graminearum* inoculation; and then continuously cultured in NaHS at the pre-treatment concentrations. After continuous treatment for 3 days, the treatment was stopped, and sterile distilled water was used to irrigate the wheat. The optimal NaHS concentration, that effectively relived the FHB symptoms, was determined according to the disease index combined with growth indexes in wheat seedlings. To confirm the hypothesis that NaHS-mediated disease suppression may be attributable to the production of $H_2S$ or $HS^-$, various sodium salts were used as controls for $H_2S$ donors, in accordance with the method described by *Zhang et al. (2008)* with some modifications. Wheat was treated in the same way as in the aforementioned experiment with water (CK), 0.3 mM NaHS (the optimal concentration), $Na_2SO_4$, $Na_2SO_3$, $NaHSO_4$, $NaHSO_3$, or NaAC. All measurements were collected with three replicates.

## Disease assessment

On 7 days post-inoculation (dpi), visible necrosis symptoms induced by *F. graminearum* on the wheat seedling stems were scored using the disease scoring system described by *Nicholson et al. (1998)*. Each disease index assessement was performed by taking 15 coleoptiles per replicate.

Lesion colour grade:

Grade 0: no disease.

Grade 1: very slight brown necrosis.

Grade 2: slight/moderate brown necrosis.

Grade 3: extensive brown necrosis.

Grade 4: extensive black necrosis.

Disease index (DI) $= \frac{\sum(lesion\ colour\ grade \times lesion\ length)}{total\ numbles\ of\ plant}$

Note: lesion length (mm).

## Determination of the wheat growth index

Wheat growth indexes were determined to evaluate the effects of different concentrations of NaHS on wheat growth. Each growth index was assessed by taking 15 seedlings per replicate. Subsequently, the dry weights of the samples were measured after 72 h in an oven at 65 °C.

## Relative quantification of *F. graminearum* DNA on the inoculated stem base of wheat

A real-time quantitative PCR (RT-qPCR) assay was performed for relative quantification of *F. graminearum* based on specific primers such as Fg16F/Fg16R according to *Nicholson et al. (1998)*. The inoculated plant sheath and frozen mycelium (PDA: potato dextrose agar, control) were finely ground with a grinder for 2 min to a fine powder. Then, fungal DNA was extracted from 100 mg of samples with the HP Fungal DNA Mini Kit (Omega, Shanghai, China).

Before qPCR, the quality of the qPCR primers was evaluated by PCR to determine whether they could yield target fragment and if the product was unique. qPCR was performed on a Roche LighterCycler 96 qPCR machine based on the protocol previously described by *Tang et al. (2020)*.

## Measurement of endogenous H$_2$S in wheat

H$_2$S content data were collected as previously described by *Sekiya et al. (1982)* with some modifications. Wheat leaves (0.3 g) were sampled and snap-frozen in liquid nitrogen. The samples were ground and treated with 1.5 mL of Tris hydrochloride (Tris–HCl) buffer (pH 8.0), after which the supernatant was collected and mixed with a zinc acetate trap in a test tube after centrifugation. After adding 0.1 mL of 20 mM dimethyl-p-phenylenediamine dissolved in 7.2 M HCl, the test tube was sealed quickly. Then, 0.1 mL of 30 mM ferric chloride in 1.2 M HCl was injected into the test tube, and the mixture was incubated for 30 min at 25 °C. Measurement of H$_2$S was performed using a UV–VIS spectrophotometer (ScanDrop, Analytikjena, Germany) at a wavelength of 667 nm. The blank was prepared with unused zinc acetate solution by using the same procedures.

## Determination of the H$_2$O$_2$ and MDA contents

The hydrogen peroxide (H$_2$O$_2$) and malondialdehyde (MDA) contents were measured in accordance with the instructions of the H$_2$O$_2$ kit and MDA assay kit (Jiancheng Bioengineering Institute, Nanjing, China), respectively.

## Measurement of SOD, CAT, APX, and POD activities

The activities of catalase (CAT), superoxide dismutase (SOD), ascorbate peroxidase (APX), and guaiacol-dependent peroxidase (POD) were assayed by the spectrophotometric method according to the kits provided by Jiancheng Bioengineering Institute.

### RNA isolation and RT-qPCR

Total RNA was isolated from samples according to the instructions provided with the Plant RNA Rapid Extraction Kit (Mei5 Biotechnology, Beijing, China). The quantity and quality of total RNA were tested by gel electrophoresis and spectrophotometry. Genes selected for further analysis included *PR1.1*, *PR2*, *PR3*, and *PR4*, with the internal control being the glyceraldehyde-3-phosphate dehydrogenase gene from *T. aestivum* (*TaGAPDH*). The primer sequences are listed in Table S1. All procedures for RT-qPCR were conducted as mentioned above.

### Evaluation of $H_2S$- induced FHB resistance in wheat heads

To further evaluate the FHB resistance regulation by $H_2S$ in wheat, wheat plants at the flowering stage were selected and transplanted under controlled environmental conditions. The wheat was subjected to the treatment described above with an optimum concentration of NaHS (0.3 mM). During this period, four spikes per plant were inoculated with 10 μL of spore suspension ($3.2 \times 10^7$ conidia/mL) of *F. graminearum* by using a syringe. Each treatment was repeated with four spikes, represent disease phenotype of wheat spikes were observed at 4 dpi.

### Effect of $H_2S$ on fungal growth

The spore suspension of *F. graminearum* was prepared in accordance with the methods described above. Four aliquots of the spore suspension (4 μL) were placed on 9-cm diameter Petri dishes maintained in sealed 1-L containers (*Hu et al., 2014*). NaHS solutions, at concentrations of 0, 0.01, 0.075, 0.1, and 0.3 mM were used to fumigate *F. graminearum* at relative humidity of 90%–95% at the bottom of the sealed containers. Spore germination (24 h) and hyphal diameter (3 d) were counted according to the method described by *Fu et al. (2014)*.

### Statistical analysis

All data were analysed by IBM SPSS 19.0 and graphed with Origin 8.5. The experimental data values, which were expressed as the mean $\pm$ standard error of the mean, were the average of the measurements obtained with triplicate independent assays. The data were evaluated by one-way ANOVA at a 95% confidence level followed by Tukey's test, where differences were considered significant at $P < 0.05$. The data collected from RT-qPCR were analysed using the $2^{-\Delta\Delta CT}$ method.

## RESULTS

### Induction of endogenous $H_2S$ content after infection with *F. graminearum*

To explore the role of endogenous $H_2S$ in the response to *F. graminearum*, the endogenous $H_2S$ content in wheat was examined. After infection with *F. graminearum*, the endogenous $H_2S$ content increased quickly in comparison with that in the control, and the $H_2S$ content in infected wheat at 12 h was approximately twice that in the control ($p < 0.05$), and it peaked at 24 h (Fig. 1). As the disease symptoms increased, the $H_2S$ content in wheat decreased. This result showed that endogenous $H_2S$ was strongly induced by
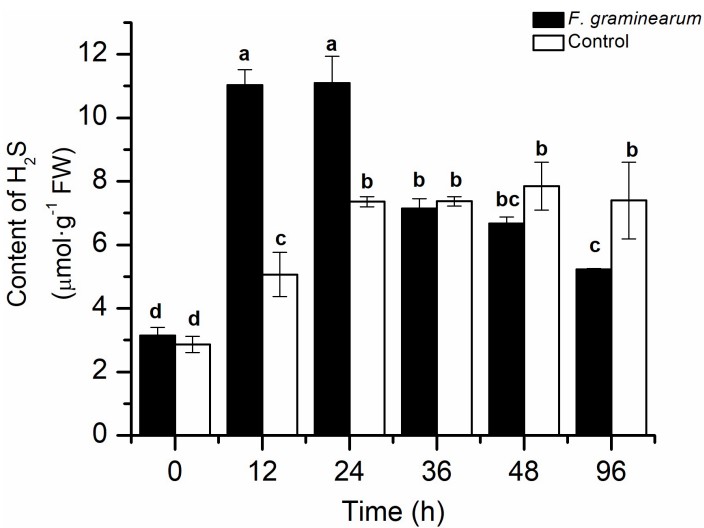

**Figure 1** **The content of H$_2$S in wheat seedlings at different time points (0, 12, 24, 36, 48, and 96 h) after infected with *F. graminearum*.** Wheat uninfected with *F. graminearum* served as control group. Inoculation was administered to two-day-old wheat. Data represent means ± S.D. of three replicate samples. Different letters mean significance of difference between the treatments according to Tukey's multiple range test at *P* < 0.05.

*F. graminearum*, and might play an important role in resistance to *F. graminearum* in wheat.

## Effects of NaHS treatments on the resistance of wheat to *F. graminearum*

To determine the toxic effects of the H$_2$S donor NaHS on wheat growth under normal conditions, wheat was watered with solutions containing 0, 0.1, 0.3, 0.5, and 1.0 mM NaHS for 3 d. As shown in Table 1, low concentrations of NaHS (0–0.5 mM) had no significant effect on the growth parameters of wheat, whereas the 1.0 mM NaHS concentration was observed to inhibit wheat growth to a certain extent, especially root length.

To explore the effect of external application of H$_2$S on FHB pathogenicity, a standard method involving inoculation of seedlings (wheat coleoptiles), which has been proven to be a reliable prescreening method for FHB resistance (*Li et al., 2019*; *Shin et al., 2014*; *Soresi et al., 2015*; *Wu et al., 2005*), was used to assess FHB severity. Photographs of representative wheat seedlings on 7 dpi were taken (Fig. 2A). After coleoptile inoculation, the leaves of plants without NaHS application turned yellow with observable fungal hyphae. In addition, a large number of hyphae and brown lesions could be observed in the diseased stems. Figure 2A showed that the application of NaHS at different concentrations (0.0, 0.1, 0.3, and 0.5 mM) reduced the disease symptoms to varying degrees. In comparison with the untreated plants, those treated with 0.5 mM NaHS subgroup showed the maximum decrease (85.85%) in the disease index (*p* < 0.001), followed by an 85.60% decline in the 0.3 mM NaHS subgroup (*p* < 0.001) (Fig. 2B). Furthermore, RT-qPCR analysis showed a significantly lower quantity of *F. graminearum* in the sheath of the NaHS/infected groups

**Table 1 Effects of NaHS treatment on wheat growth index under normal conditions.**

| Concentration of NaHS treatment (mM) | 0.0 | 0.1 | 0.3 | 0.5 | 1.0 |
|---|---|---|---|---|---|
| Plant height (cm) | 12.033 ab (100) | 12.544 a (104) | 12.402 a (103) | 11.969 ab (99.5) | 11.120 b (92.4) |
| Root length (cm) | 16.591 a (100) | 16.278 a (98.1) | 16.313 a (98.3) | 14.313 a (86.3) | 11.574 b (69.8) |
| Fresh weight (mg/plant) | 260.484 a (100) | 265.790 a (102) | 268.873 a (103) | 258.920 a (99.4) | 229.436 b (88.1) |
| Dry weight (mg/plant) | 31.456 a (100) | 32.900 a (104) | 33.209 a (105) | 33.633 a (107) | 31.553 a (100) |

**Notes.**

Two-day-old wheat seedlings were culture in 0.0 (control), 0.1, 0.3, 0.5, and 1.0 mM NaHS respectively for 3 days, then the growth index was investigated on 9 dps (days post sowing). The data are presented as the mean ± SD ($n = 15$); a, b and c mean significant differences at $P < 0.05$ for the same row; The data shown in brackets is the percentages of the control group.

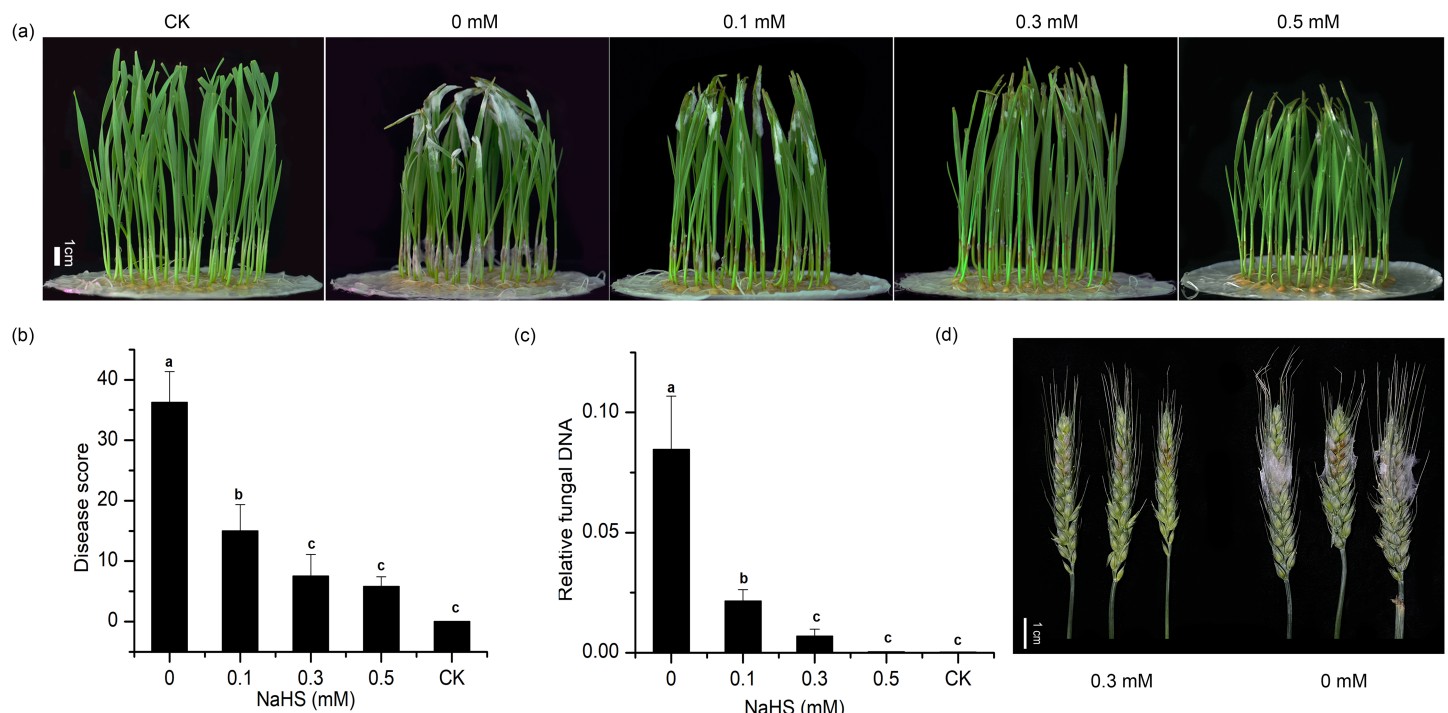

**Figure 2 Effects of different concentrations of NaHS treatment on wheat disease resistance to *F. graminearum*.** (A) Disease phenotype of wheat seedlings on 7 dpi. (B) Disease score of wheat seedlings on 7 dpi. (C) qPCR analysis showing levels of relative fungal DNA in wheat seedlings. (D) Disease phenotype of wheat spike at 4 dpi. CK:healthy control that uninfected with *F. graminearum*. Data represent means ± S.D. of three replicate samples. Different letters mean significance of difference between the treatments according to Tukey's multiple range test at $P < 0.05$.

in comparison with the non-NaHS/infected group (Fig. 2C), especially at concentrations of either 0.3 or 0.5 mM ($p < 0.01$). These findings suggest that NaHS could ameliorate the negative effects of *F. graminearum* stress.

As shown in Table 2, the non-NaHS/infected group showed a notable decrease in plant height, fresh weight and dry weight, whereas exogenous NaHS inhibited the decline in wheat seedling growth indexes under *F. graminearum* stress. Moreover, an increase in the concentration of NaHS from 0.3 mM to 0.5 mM led to reductions in plant height, root

**Table 2  Effect of exogenous NaHS on wheat growth parameters under stressed conditions.**

| Concentration of NaHS treatment (mM) | CK | 0.0 | 0.1 | 0.3 | 0.5 |
|---|---|---|---|---|---|
| Plant height (cm) | 12.103 a (100) | 10.827 b (89.5) | 11.173 ab (92.3) | 11.357 ab (93.8) | 10.893 b (90) |
| Root length (cm) | 12.627 a (100) | 11.400 a (90.3) | 12.077 a (95.6) | 11.903 a (94.3) | 11.357 a (90) |
| Fresh weight (mg/plant) | 283.109 ab (100) | 256.769 b (90.7) | 283.251 ab (100) | 288.367 a (101.9) | 274.209 ab (96.9) |
| Dry weight (mg/plant) | 32.920 a (100) | 28.949 b (87.9) | 30.760 ab (93.4) | 31.387 ab (95.3) | 31.696 ab (96.3) |

Notes.

The growth index was investigated on 7 dpi. CK:healthy control. The data are presented as the mean ± SD ($n = 15$); a, b and c mean significant differences at $P < 0.05$ for the same row; The data shown in brackets is the percentages of the control group.

length, and fresh weight. On the basis of these results, we chose a working concentration of 0.3 mM NaHS for the subsequent analyses.

To further verify the resistance regulation of $H_2S$ to FHB in wheat, spike inoculated wheat was treated with 0.3 mM NaHS (Fig. 2D). The result showed that NaHS significantly reduced FHB severity in spike, which was consistent with the phenomenon in coleoptile. On the basis of these results, we could hypothesise that the application of NaHS may confer resistance to FHB in wheat and result in suppression of fungal propagation.

## Protective roles of $H_2S$ in amelioration of *F. graminearum* stress in wheat

To investigate the effect of $H_2S$ in alleviating disease symptoms induced by NaHS in wheat inoculated with *F. graminearum*, 0.3 mM $Na_2SO_4$, $Na_2SO_3$, $NaHSO_4$, $NaHSO_3$, and NaAC were applied in the same experimental system. As shown in the data in Fig. 3A, the control groups with $Na^+$ or sulphur-containing components did not show reduced the disease severity as the NaHS group did, which indicated that $H_2S$ or $HS^-$, rather than other derivatives, promoted the role of NaHS in heightening FHB resistance in wheat.

The endogenous $H_2S$ level in wheat seedlings was measured under various treatment conditions (Fig. 3B). Marked modulation of endogenous $H_2S$ content in the results showed that NaHS pre-treatment caused a change in the $H_2S$ content of wheat seedlings. Briefly, the $H_2S$ content in pre-treated plants was higher than that the control plants within 36 h, especially at 6′/0 h and 9 h, and almost tripled in comparison with the values in the control group ($p < 0.001$). Meanwhile, the time-course of changes in the endogenous $H_2S$ content in the control group in response to *F. graminearum* was consistent with the results in Fig. 1. These results showed that the increase in endogenous $H_2S$ levels through exogenous application of NaHS could contribute to resistance against *F. graminearum* in wheat seedlings.

## Exogenous $H_2S$ modulated oxidative stress in wheat seedlings under biotic stress conditions

To further understand the mechanism underlying $H_2S$-induced resistance against *F. graminearum*, the $H_2O_2$ and MDA contents were determined. An $H_2O_2$ burst occurred in the control group after infection, while the $H_2O_2$ content was notably reduced in the NaHS group (Fig. 4A). The $H_2O_2$ levels increased by approximately 59.6% in the control
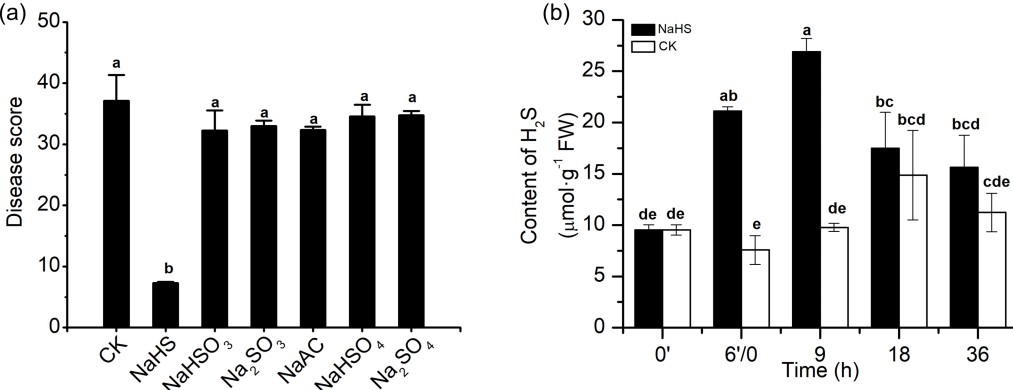

**Figure 3 Effects of H₂S on wheat resistance to *F. graminearum* through NaHS treatment.** (A) The protective roles of NaHS on amelioration of FHB stress in wheat could be attributed to $H_2S$ or $HS^-$. Seedlings were pretreated with $H_2O$ (CK), 0.3 mM NaHS, $Na_2SO_4$, $Na_2SO_3$, $NaHSO_4$, $NaHSO_3$, and NaAC, respectively for 6 h, and subsequently subjected to FHB stress, then watered after 3 days of each compound incubation, disease index on 7 dpi was investigated. (B) Effect of NaHS treatment on endogenous $H_2S$ content in wheat seedlings exposed to FHB stress. After 6 h of pretreatment, seedlings were inoculated with *F. graminearum*. 0′ represented seedlings sample without NaHS treatment; 6′/0 represented the samples from the seedlings pretreated with water (Control, CK) or NaHS (0.3 mM) after 6 h, and at the time point, subjected to inoculation with *F. graminearum*; and 9, 18 and 36 as samples of the seedlings inoculated with *F. graminearum* for 9, 18 and 36 h respectively. Data represent means ± S.D. of three replicate samples. Different letters mean significance of difference between the treatments according to Tukey's multiple range test at $P < 0.05$.

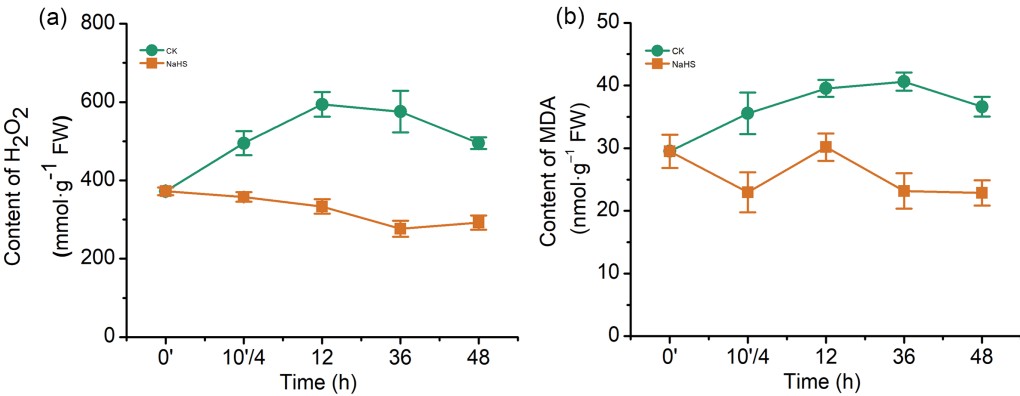

**Figure 4 Effect of H₂S on the contents of hydrogen ($H_2O_2$) (A) and malonaldehyde (MDA) (B) in wheat seedlings.** 0′ represented seedlings sample without NaHS treatment; 10′/4 represented the samples from the seedlings pretreated with water (Control, CK) or NaHS (0.3 mM) after 10 h, and at the time point, 4 h post inoculation; and 12, 36 and 48 as samples of the seedlings inoculated with *F. graminearum* for 12, 36 and 48 respectively. Data represent means ± S.D. of three replicate samples.

group ($p < 0.001$), whereas NaHS treatment decreased the $H_2O_2$ content by 10.4% after 12 h of inoculation ($p < 0.05$).

MDA, a product of membrane lipid peroxidation, is one of the most commonly used indexes of lipid peroxidation. The MDA content in the control wheat samples increased

significantly up to 36 h of inoculation, which was twice that of the treatment group ($p < 0.001$), and subsequently decreased (Fig. 4B). In addition, it was lower in the NaHS group than in the control group during the entire treatment period. These results indicated that *F. graminearum* could cause $H_2O_2$ accumulation and membrane damage in wheat seedlings, while exogenous application of $H_2S$ could relieve the oxidative stress caused by $H_2O_2$ and MDA.

### Effects of $H_2S$ on the activities of the antioxidant enzymes in wheat grown under biotic stress

The major antioxidant enzymes include SOD, POD, APX, and CAT, which detoxify reactive oxygen species (ROS) in various plant tissues. In our experiments, the activities of antioxidant enzymes (SOD, POD, APX, and CAT) in the NaHS group were substantially higher than those in the control group during most of the treatment period (Fig. 5). SOD activity in the NaHS group and control group increased sharply and reached the maximum level at 48 h, which was 22.09% higher than that in the control ($p < 0.01$), and then rapidly declined from 48 h to 72 h (Fig. 5A). Additionally, NaHS treatment activated POD activity, which was approximately 1.34 times higher than that in the control group at 36 h ($p < 0.01$). APX activity varied similarly in the control and NaHS groups, while APX activity in the NaHS group was higher than that in the control group during the entire treatment period ($10'/4$ and 36 h, $p < 0.05$), except for 72 h (Fig. 5C). Furthermore, there was no obvious difference in CAT activity between the NaHS and control groups at the beginning. This gap narrowed from 0 to 12 h, after which CAT activity in the NaHS group increased faster than that in the control group, and was 1.71 times the activity in the control group at 72 h ($p < 0.01$) (Fig. 5D). Unambiguously, these results demonstrated that alleviation of oxidative stress by $H_2S$ in wheat under FHB stress was closely related to improvement of antioxidant system activities in wheat seedlings.

### Activation effects of $H_2S$ on defence response genes

Defence response genes, including genes encoding pathogenesis-related proteins (*PRs*), are involved in the resistance response to many plant-pathogen interactions (*Hammond-Kosack & Jones, 1996*; *Pritsch et al., 2000*; *Pritsch et al., 2001*; *Stevens et al., 1996*). To examine the association between $H_2S$-induced resistance and *PR* gene expression in wheat, we conducted a quantitative analysis for the *PR1.1*, *PR2*, *PR3* and *PR4* genes after NaHS application (Fig. 6). The results showed that $H_2S$ upregulated the transcription levels of *PR1.1*, *PR2*, *PR3* and *PR4*, which peaked at 12 h and 48 h, with the degree of between 2 and 4 times (*PR1.1*, *PR2*, and *PR3*, peaked at 12 h, $p < 0.001$; *PR4*, peaked at 48 h, $p < 0.05$). Moreover, the relative expression levels of *PRs* in the NaHS group were higher than those in the control group within 48 h. Thus, our results demonstrated that exogenous $H_2S$ improved disease resistance against *F. graminearum* by advancing the accumulation of defence-related genes.

### Effects of $H_2S$ on mycelial growth and spore germination

To further understand the mechanism underlying the regulatory effects of $H_2S$ in resistance to *F. graminearum* in wheat, the effects of $H_2S$ on mycelial growth and spore germination were investigated (Fig. 7). As shown in Fig. 7, with increasing NaHS concentration, the

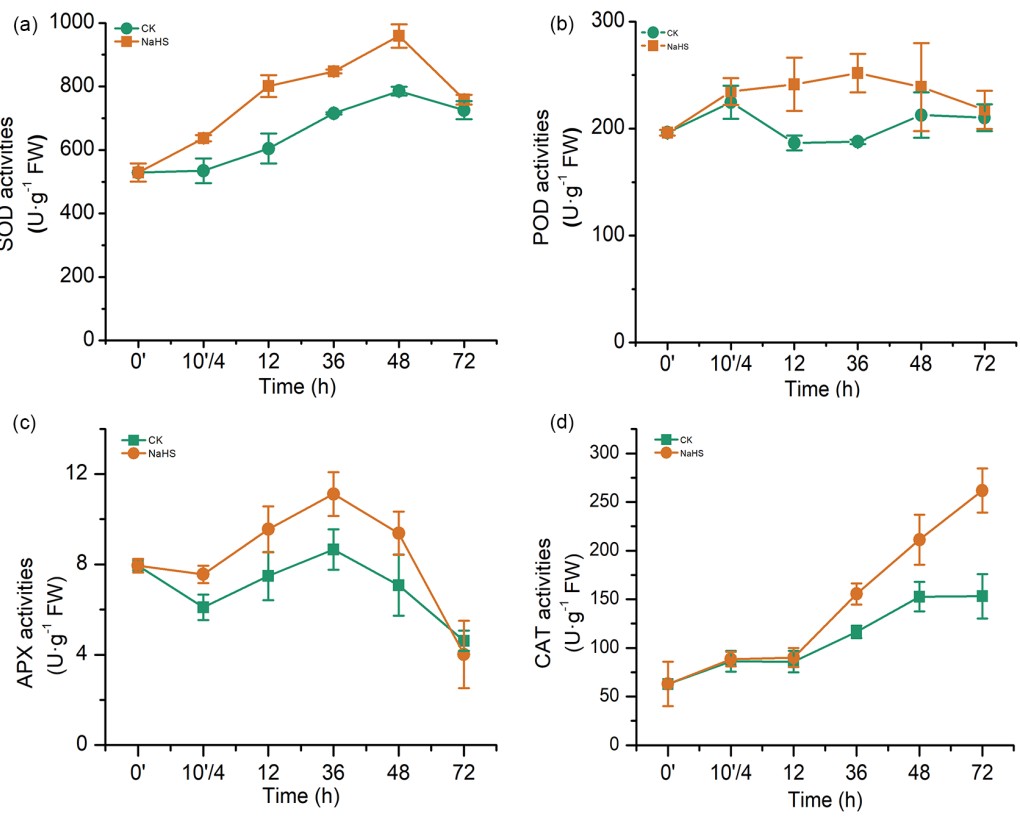

**Figure 5** Effects of H$_2$S on activities of antioxidant enzymes (A) SOD activity, (B) POD activity, (C) APX activity, and (D) CAT activity. 0' represented seedlings sample without NaHS treatment; 10'/4 represented the samples from the seedlings pretreated with water (Control, CK) or NaHS (0.3 mM) after 10 h, and at the time point, 4 h post inoculation; and 12, 36, 48, and 72 as samples of the seedlings inoculated with *F. graminearum* for 12, 36, 48, and 72 h respectively. Data represent means ± S.D. of three replicate samples.

mycelial diameter of *F. graminearum* on PDA decreased, and mycelial growth was inhibited by 37.66% even at 0.075 mM NaHS treatment ($p < 0.001$) (Fig. 7B). At 0.3 mM NaHS, no signs of mycelial growth were observed. Moreover, in our assessment of spore germination after 24-h incubation, and most of the fungal spores germinated at 0.01 mM. However, treatment with 0.1 mM NaHS completely restrained spore germination of *F. graminearum* (Figs. 7A, 7C). These data showed that exposure to H$_2$S notably inhibited mycelial growth and spore germination of *F. graminearum* in a dose-dependent manner, suggesting that H$_2$S had an effectively fungicidal role.

## DISCUSSION

### Application of exogenous H$_2$S reduced FHB symptoms in wheat

*Fusarium* diseases resulted in severe economic losses for the production of cereal crops. Global climate change has increased the contribution of weather conditions to epidemics of wheat FHB, and control of this disease has proven to be exceedingly challenging (*Zhang et al., 2014*). H$_2$S is an essential signalling molecule that exhibits positive effects on

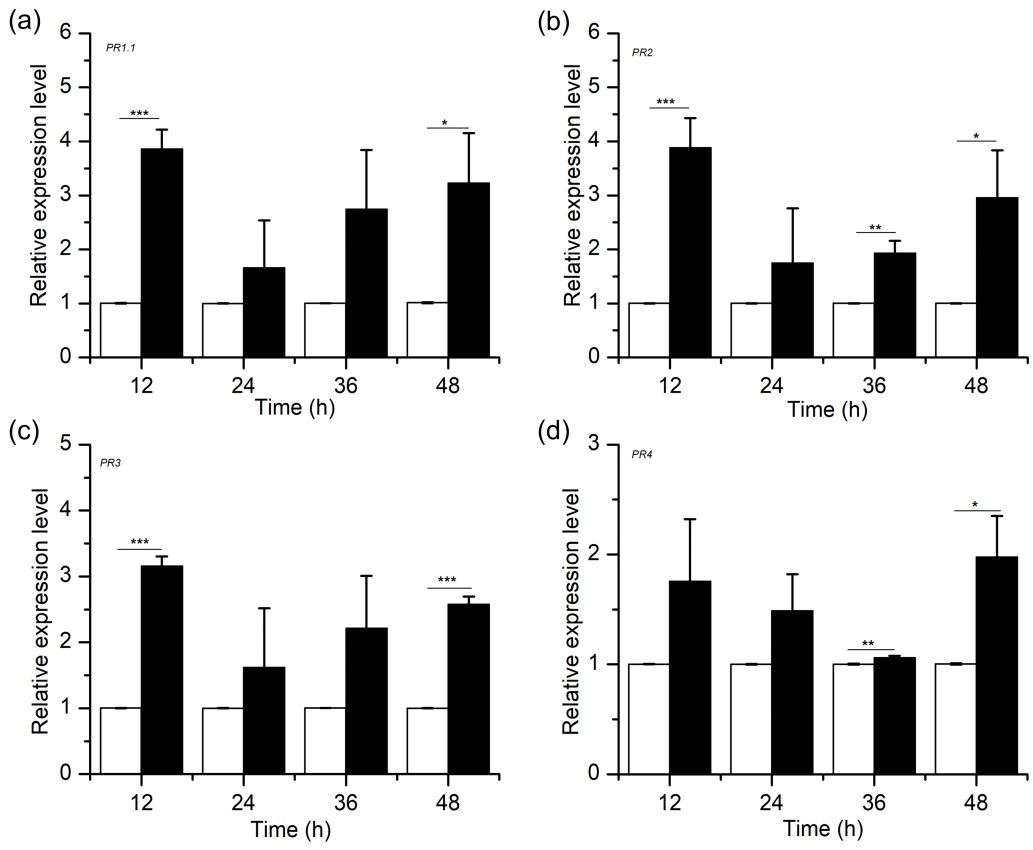

**Figure 6** **Effects of H₂S treatment on defense-related gene (A)** ***PR1.1,*** **(B)** ***PR2,*** **(C)** ***PR3,*** **(D)** ***PR4.*** Significant differences between 0.3 mM NaHS treatment group and control group were compared by Tukey's multiple range test. Significant difference at $P < 0.001$, $P < 0.01$ and $P < 0.05$ were marked as triple (***), double (**) and single (*), respectively. Data represent means ± S.D. of three replicate samples.

numerous physiological functions, including those serving against environmental stress and pathogens. Treatments of H₂S combined with silicon have been reported to alleviate the negative effects of drought stress and rust infection in wheat (*Naz et al., 2021*). Our study demonstrated that exogenous H₂S alleviated FHB symptom in wheat seedlings, enhancing the plant defense mechanism and inhibiting fungal growth.

In this study, H₂S content was conspicuously induced after *F. graminearum* infection, suggesting that H₂S may be partly responsible for the defence response to *F. graminearum* in wheat (Fig. 1). This result was supported by previous studies in *Arabidopsis* showing that the endogenous H₂S content increased significantly after pathogen infection (*Shi et al., 2015*). Nevertheless, previous to this study, the role of H₂S in wheat resistance to FHB was not clear.

Coleoptile inoculation method can be used as a rapid and reliable method for FHB pathogenicity research, and it has been described in many studies in the literature (*Wu et al., 2005*; *Yang et al., 2018*). Comparative pathogenic analyses indicated that the application of NaHS effectively reduced the disease symptoms of wheat seedlings in a dose-dependent

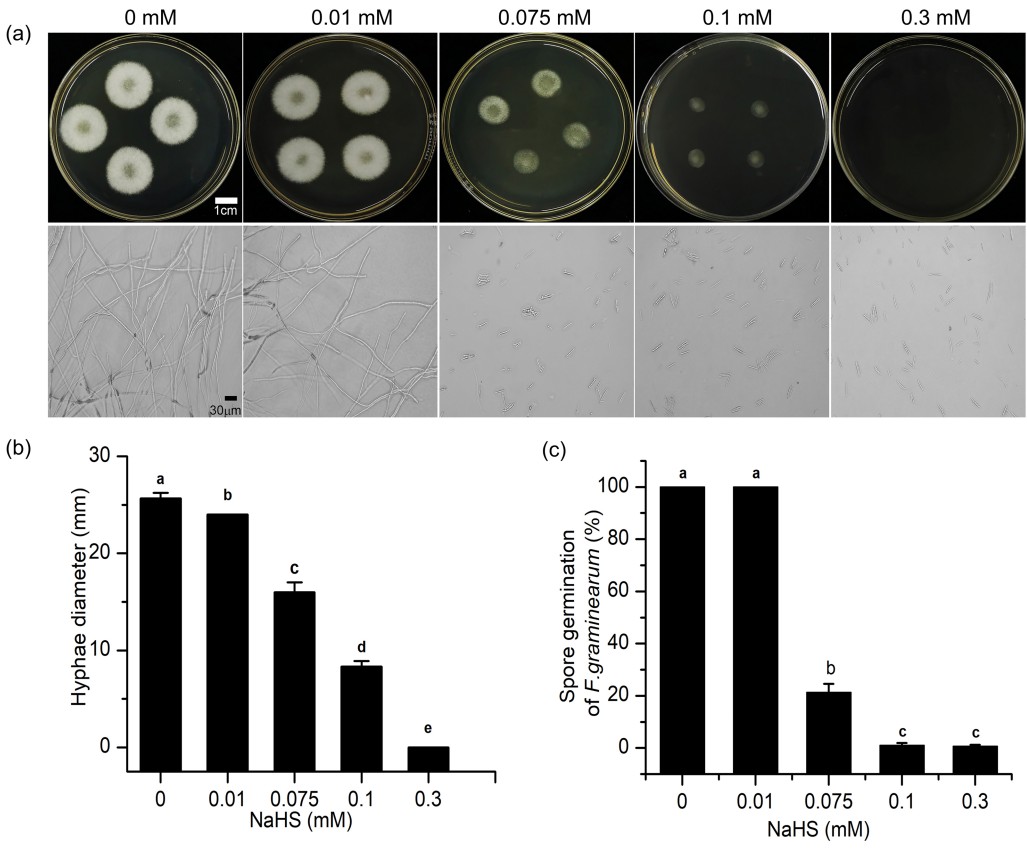

**Figure 7** H$_2$S effects development of *F. graminearum* on potato dextrose agar (PDA) (A) Mycelium growth of *F. graminearum*, which were cultured on medium and subjected to different concentrations of NaHS solutions for 3 days. Lower panel represents micro-morphology of *F. graminearum*, which were cultured on medium and subjected to different concentrations of NaHS solutions for 24 h. (B) Hyphae diameter of fungi clones (culture for 3 d). (C) Spore germination of *F. graminearum* (culture 24 h). Data represent means ± S.D. of three replicate samples. Different letters mean significance of difference between the treatments according to Tukey's multiple range test at $P < 0.05$.

manner, and at an optimal concentration of 0.3 mM, the FHB severity of wheat spikes was also alleviated (Fig. 2). NaHS is a commonly used H$_2$S donor in biological systems that decomposes into Na$^+$ and HS$^-$ in solution, of which HS$^-$ combines with H$^+$ to generate H$_2$S (*Zhang et al., 2010a*; *Zhang et al., 2010b*). The role of NaHS as an H$_2$S donor, in reducing the negative effects of *F. graminearum* in wheat was supported by two sets of evidence. First, only NaHS could effectively reduce the FHB severity of wheat in the solutions we tested. Moreover, as shown in Table 1, NaHS itself did not negatively affect seedling growth at low concentrations, implying that NaHS counteracts the pathogen-induced stresses in plants in a way that was independent of any "cross-tolerance". On the other hand, wheat seedlings treated with NaHS showed markedly higher the levels of extractable H$_2$S compared with the control (Fig. 3). These data provided new insights, hinting at a possible role of H$_2$S in the plant defence signalling of wheat against *F. graminearum*.

## H$_2$S enhanced the resistance of wheat to *F. graminearum* by enhancing the defense mechanism

ROS, especially H$_2$O$_2$, are known to be important in plant-pathogen interactions. Low levels of H$_2$O$_2$ are essential for peroxidase-dependent lignification, which hinders penetration by a pathogen. However, necrotrophic pathogens induced H$_2$O$_2$ at toxic levels, which could lead to cell death, and the pathogens could use these dead cells as an entry point into plants or obtain nutrients from dead cells for further invasion and growth (*Hückelhoven, 2007*). Correspondingly, during the process of infection and colonisation of grain hosts, *Fusarium* causes oxidative stress in the host by secreting toxic levels of ROS including H$_2$O$_2$, which acts as weapons to attack plants (*Walter, Nicholson & Doohan, 2010*). In this study, *F. graminearum*-infected wheat showed a marked increase in H$_2$O$_2$ levels, and this increase was mitigated by treatment with NaHS (Fig. 4). Furthermore, considering the MDA content, control plants showed elevated MDA levels after infection, which was likely connected with the remarkable accumulation of H$_2$O$_2$, implied that rapid oxidative explosion destroyed membranes. Plants receiving NaHS treatment showed significantly lower post-infection MDA levels in comparison with the control group, indicating decreased ROS levels, and a reduced degree of membrane peroxidation. Taken together, these data support the idea that H$_2$S appears to be an antioxidant signalling molecule participating in the resistance against *F. graminearum* in wheat.

To further examine the role of NaHS in relieving ROS toxicity, the activities of various antioxidant enzymes, including SOD, POD, APX, and CAT, were evaluated. As shown in Fig. 5, the activities of SOD, POD, APX, and CAT were upgraded by NaHS treatment, indicating that H$_2$S could enhance the activities of antioxidant enzymes to scavenge ROS, which was consistent with findings showing H$_2$S attenuated the rot of freshly cut pears by enhancing the antioxidant system to exterminate excessive ROS (*Hu et al., 2014*). *In vitro* assays showed that APX and CAT were targets of persulfidation, which were positively modulated by persulfidation in the presence of H$_2$S (*Corpas, 2019*). Moreover, exogenous H$_2$S treatment has also been shown to relieve the ROS burst and cell damage induced by drought, salinity, temperature, and heavy metals in plants (*Corpas, 2019*). The present data also speculated that the resistance of the FHB-resistant variety 'Vulkan' was associated with antioxidative enzyme activity (*Spanic et al., 2017*). These results may partly account for the effects of H$_2$S treatment in inducing resistance to *F. graminearum* in wheat.

*PRs* are related to the development of systemic acquired resistance (SAR) against further infection enforced by pathogeny. Additionally, published papers have described that the genes encoding *PRs* in wheat were activated after *F. graminearum* attack (*Pritsch et al., 2000*; *Pritsch et al., 2001*). However, physiological concentrations of H$_2$S have been shown to upregulate the expression levels of disease-related genes (*Shi et al., 2015*). For example, the auxin signalling pathway mediated by MIR393 includes MIR393a/b and its target genes (*TIR1*, *AFB1*, *AFB2*, and *AFB3*), which were all transcriptionally regulated by H$_2$S (*Shi et al., 2015*). In this study, we noted that, the *PR* gene transcript accumulation in NaHS-treated wheat was greater than that in control plants (Figs. 6A–6D). Taken together, these results could indicate that H$_2$S displayed the ability to enhance the expression of defence response genes, thereby possibly contributing to greater disease resistance in wheat.

### The effects of H₂S on the growth of *F. graminearum in vitro*

We also studied the toxicity of $H_2S$ to *F. graminearum*, and found that $H_2S$ released by 0.3 mM NaHS hindered the development of *F. graminearum* during the testing period (Figs. 7A–7C). Our results suggested that $H_2S$ exerted its antifungal effect by influencing various aspects of fungal development, including inhibition of spore germination and retardation mycelial growth. Similar effects of $H_2S$ have been confirmed by *Fu et al. (2014)* and *Hu et al. (2014)*.

### The potential application of H₂S in wheat FHB in the future

Our experiments showed that $H_2S$ could be a potential superior candidate for controlling FHB. NaHS has been widely used as the rapid $H_2S$ donor that generates a large amount of $H_2S$ over a short time period (*Sun et al., 2017*). The rapid generation and loss of $H_2S$ limit its application in agriculture. Controlled release of $H_2S$ could make it safer and more cost-effective to use. With this in mind, *Lee et al. (2011)* previously reported that the slow-releasing $H_2S$ donor, GYY4137, exhibited novel anti-cancer effects *in vitro* and *in vivo*. Likewise, in the work of *Liu et al. (2021)*, they provided an infrared-light-responsive pesticide delivery system to intentionally regulate imidacloprid release and enhance utilization efficiency. Hence, it would be valuable to develop a delivery system that could be manipulated to achieve sustained release of $H_2S$ from its donor NaHS. In the future, reformulated NaHS could be applied to plants or sprayed directly into the soil to inhibit the fungal, which might have an application value.

## CONCLUSIONS

In conclusion, irrespective of the inoculation methods, the application of $H_2S$ released by 0.3 mM NaHS could significantly alleviate FHB symptoms in wheat. $H_2S$ enhanced the resistance of wheat to *F. graminearum* by enhancing the defence mechanism (i) promotion of the activities of antioxidant enzymes with decreasing the amount of $H_2O_2$ and MDA and (ii) induction of the expression of disease response genes. On the other hand, $H_2S$ was capable of inhibiting the development of the pathogen. On the basis of these data, we have provided a theoretical basis for $H_2S$-mediated resistance to *F. graminearum* in wheat.

**Abbreviations**

| | |
|---|---|
| **RT-qPCR** | real-time quantitative PCR |
| **H₂S** | hydrogen sulfide |
| **MDA** | malondialdehyde |
| **FHB** | *Fusarium* head blight |
| ***F. graminearum*** | *Fusarium graminearum* |
| **ROS** | reactive oxygen species |
| **H₂O₂** | hydrogen peroxide |
| **SA** | salicylic acid |
| **PRs** | pathogenesis-related proteins |
| **CMC** | Carboxymethyl cellulose |
| **NaHS** | sodium hydrosulfide |

| mM | mmol/L |
|---|---|
| DI | disease index |
| SOD | superoxide dismutase |
| CAT | catalase |
| PDA | potato dextrose agar PDA |
| dps | days post sowing |
| dpi | days post inoculation |
| APX | ascorbate peroxidase |
| POD | peroxidase |

## ACKNOWLEDGEMENTS

We are grateful to Xian Shu, Chinese Academy of Sciences for his suggestions and efforts in revising of the manuscript.

### Funding

This work was supported by the Major Special Project of Anhui Province, China (No.202103a06020002) and the Grant of the President Foundation of Hefei Institutes of Physical Science of Chinese Academy of Sciences, China (No.YZJJ2020QN29). The funders had no role in study design, data collection and analysis, decision to publish, or preparation of the manuscript.

### Grant Disclosures

The following grant information was disclosed by the authors:
Major Special Project of Anhui Province, China: No.202103a06020002.
The Grant of the President Foundation of Hefei Institutes of Physical Science of Chinese Academy of Sciences, China: No.YZJJ2020QN29.

### Competing Interests

The authors declare there are no competing interests.

### Author Contributions

- Yuanyuan Yao and Caiguo Tang conceived and designed the experiments, performed the experiments, analyzed the data, prepared figures and/or tables, authored or reviewed drafts of the paper, and approved the final draft.
- Wenjie Kan and Pengfei Su performed the experiments, analyzed the data, prepared figures and/or tables, and approved the final draft.
- Yan Zhu, Wenling Zhong, Jinfeng Xi and Dacheng Wang analyzed the data, prepared figures and/or tables, and approved the final draft.
- Lifang Wu conceived and designed the experiments, authored or reviewed drafts of the paper, and approved the final draft.

## Data Availability

The raw measurements are available in the Supplemental Files.

## Supplemental Information

Supplemental information for this article can be found online at http://dx.doi.org/10.7717/peerj.13078#supplemental-information.

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
