# Peer review of "Hydrogen sulphide alleviates Fusarium Head Blight in wheat seedlings"

_PeerJ, doi:10.7717/peerj.13078_

## Round 0.1 · original submission · Major Revisions

There are good comments from the three independent reviewers on your article. I think all the queries are addressable by the authors of the manuscript. Please see the comments and respond to them one by one.

Reviewer 1 ·

Basic reporting

The manuscript was well written in general.

In introduction, review of many others' papers was not cited. Please add relevant literature citations when review others' reports.

Experimental design

Critical methods were not clearly described.
1. Was the experiment replicated? FHB is a difficult disease to be studied and usually people need conduct several experiments to repeat phenotypic data.
2. How the spike was inoculated? One spikelet per spike? How many spikes per treatment? How many rep? Were these inoculated spikes covered? If so how many days? These are important information for readers to justify if the experimental data support the conclusions or not, but they were missing in the manuscript.

Validity of the findings

The finding is novel and interesting.

But without detailed description of the disease evaluation, readers have no way make judgement of validity of the findings.

Additional comments

If the finding is real, how farmer can use the finding to control FHB? Can farmers apply it with fertilizer or apply it with fungicide? Please discuss possible method to use the finding in FHB control by farmers.

Reviewer 2 ·

Basic reporting

The manuscript by Yao et al., describes the potential value of H2S in mitigating the adverse effects induced by Fusarium graminearum and advanced the current knowledge regarding the molecular mechanisms in wheat.
The manuscript is well written with relevant comprehensive Introduction, Literature and Cited references throughout the manuscript. In addition to this, figures are relevant and well labelled having high quality and comprehensive legends. Raw data is also provided.
There is need to rephrase the sentences in the following lines to make them more clear:
Lines# 142-143, 207, 232, 288-290, 313-315.
Figure1: Please rephrase the figure legend to make it clear and attractive.

Experimental design

There are some points which should be addressed by authors(s) to make them more clear:
Line # 100-102: There is a need to describe the alterations in the methodology.
Line# 109: Please provide justification for using pre-treatment for 6h and not 5/4/3h…..
Line# 109: There is mentioned pre-treatment of seedling samples @ 0, 0.1, 0.3, and 0.5 Mm WHILE Line# 182: on fungal growth, 2 different concentrations were applied @ 0.01 & 0.075. Please give justification to make it clear.
Line#203: The concentration 1.0 mM NaHS is mentioned in results but not in methodology. Why?

Validity of the findings

The manuscript by Yao et al., provides innovative as well as informative findings.
All understudying data have been provided which looks are robust, statistically sound, & controlled.
Conclusions are well stated, linked to original research question & limited to supporting results.
Anyhow, there is need to provide justification/reason why disease score is significantly lower in CK compared to 0 mM (without) NaHS concentration in Figure 2.

Additional comments

This present research work is innovative and informative. I have some observations which author(s) should need to be addressed in this study:
i) Fusarium graminearum is notorious due to DON toxin production besides impact on wheat growth related parameters, what is an impact of NaHS on DON toxin production?
ii) Please mention specific strain of Fusarium graminearum which was used in this study.
iii) Please mention the source from where authors obtained or isolated Fusarium graminearum used in this study.
iv) Please briefly describe the F. graminearum inoculum preparation method in the materials section.

Reviewer 3 ·

Basic reporting

No comments

Experimental design

Line 142:
Which house keeping genes were used as a control in qPCR as a control for detecting the relative quantity of F. graminearum on infected plants?

Line 189:
Why tukey test was used? If applied control treatment I think the best pairwise comparison test is Dunnett's test here. This must be checked because whenever you have a control treatment.

Validity of the findings

Lines 202-207
If correlation studies among biochemical traits like H2O2, POD, CAT, SOD, APX, etc. and seedling parameters or morphological traits will be added it will provide a detailed understanding of how the relative expression of antioxidants effects the wheat seedling growth and its yield related parameters.

Lines 261-263
Why not the levels of H2O2 was determined in cell organelles for detailed understanding of the pathogen induced increased level of H2O2 inside the cell.

Additional comments

No Comments

---

## Round 0.2 · accepted · Accept

All of queries are addressed in the revised version.

Reviewer 2 ·

Basic reporting

All the queries have been successfully addressed by authors and Manuscript was revised accordingly.

Experimental design

All the queries have been addressed successfully by authors.

Validity of the findings

Manuscript was successfully revised and all quires were addressed accordingly by the authors.

Additional comments

Needful has been done by authors and now this manuscript is acceptable for publication in the PEER J.